# Screening *Digitaria eriantha* cv. Suvernola Endophytic Bacteria for Maize Growth Promotion

**DOI:** 10.3390/plants12142589

**Published:** 2023-07-08

**Authors:** Michelle J. G. Alves, Johny Jesus Mendonça, Gisely Moreira Vitalino, José Paula Oliveira, Erix Xavier Carvalho, Felipe José Cury Fracetto, Giselle Gomes Monteiro Fracetto, Mario Andrade Lira Junior

**Affiliations:** 1Departamento de Agronomia, Universidade Federal Rural de Pernambuco, Recife 52171-900, Brazil; michellejustinoalves@gmail.com (M.J.G.A.); mendonca.johny@yahoo.com.br (J.J.M.); fracettofelipe@gmail.com (F.J.C.F.); giselle.fracetto@ufrpe.br (G.G.M.F.); 2Programa de Pós-Graduação em Ciência do Solo, Universidade Federal do Rio Grande do Sul, Porto Alegre 91501-970, Brazil; 3Instituto Agronômico de Pernambuco, Recife 50171-000, Brazil; jose.paula@ipa.br (J.P.O.); eric.carvalho@ipa.br (E.X.C.)

**Keywords:** biological nitrogen fixation, phosphorus solubilization, IAA, siderophores, plant growth

## Abstract

The search for sustainable agriculture has increased interest in using endophytic bacteria to reduce fertilizer use and increase stress resilience. Stress-adapted plants are a potential source of these bacteria. Some species of these plants have not yet been evaluated for this, such as pangolão grass, from which we considered endophytic bacteria as potential plant growth promoters. Bacteria from the root, colm, leaves, and rhizospheric soil were isolated, and 132 strains were evaluated for their in vitro biological nitrogen fixation, IAA and siderophores production, and phosphate solubilization. Each mechanism was also assessed under low N availability, water stress, and low-solubility Fe and P sources in maize greenhouse experiments. All strains synthesized IAA; 63 grew on N-free media, 114 synthesized siderophores, and 46 solubilized P, while 19 presented all four mechanisms. Overall, these strains had better performance than commercial inoculant in all experiments. Still, in vitro responses were not good predictors of in vivo effects, which indicates that the former should not be used for strain selection, since this could lead to not testing strains with good plant growth promotion potential. Their heterologous growth promotion in maize reinforces the potential of stress-adapted plant species as potential sources of strains for inoculants.

## 1. Introduction

Population growth demands increased food production, while current and forecasted environmental conditions require a significant decrease in agriculture’s ecological footprint. This has led to the increasing use of plant-growth-promoting bacteria (PGPB), since these can reduce fertilizer use and increase crops’ resilience to environmental stresses [1]. These are usually endophytic or rhizospheric bacteria that do not provoke any harm to the plant and may, under some conditions, promote their growth or reduce the effects of environmental stresses [2].

Endophytic bacteria may increase plant growth and adaptation to environmental stress [3]. This plant growth promotion is variously credited to several mechanisms, such as biological nitrogen fixation, synthesis and release of phytohormones such as IAA, phytopathogen control through synthesis and release of siderophores, or phosphate solubilization, several of which are frequently evaluated under in vitro conditions [4,5,6].

These mechanisms are widely distributed in several bacterial genera, such as *Acetobacter*, *Aerobacter*, *Aeromonas*, *Agrobacterium*, *Azospirillum*, *Bacillus*, *Burkholderia*, *Chryseomonas*, *Curtobacterium*, *Enterobacter*, *Erwinia*, *Flavimonas*, *Gluconacetobacter*, *Herbaspirillum*, *Klebsiella*, *Pseudomonas*, *Rhizobium*, and *Sphingomonas*, which have been isolated as endophytic from a broad spectrum of plant species, such as maize, wheat, rice, sugarcane, sorghum, and signal grass [7,8,9,10,11,12].

On the other hand, several plant species well adapted to stressful environments, such as pangolão grass (*Digitaria eriantha* cv. Survenola), have not been well studied until now, even though we have found it to harbor a very diverse bacterial endophyte population [13], since the literature contains several examples of plants from these environments being effective sources of plant-growth-promoting bacteria [14,15,16]. At the same time, it is also known that bacteria isolated from one species may be highly effective on another; for example, as happens with the commercial inoculants based on *Azospirillum brasilense* in Brazil [17]. Other example is the use of bacteria isolated from *Stipa tenacissima* L. and used to reduce salt stress on tomato [16], from *Opuntia ficus-indica* increasing wheat growth under drought [18], from garlic on common beans growth [19], or from *Suaeda nudiflora* on *Amaranthus viridis* under salt stress [3].

We aim, thus, to evaluate this diversity as a source of plant-growth-promoter bacteria both in vitro and for maize plants.

## 2. Results

### 2.1. Plant-Growth-Promoting Characteristics of Endophytic Isolates

The most frequent in vitro growth-promoting mechanism was IAA production, which occurred in all strains ranging from *ca.* 4 to *ca.* 212 μg mL^−1^, followed by siderophores (86%, ranging from 1 to 83%), BFN (48%), and phosphate solubilization (35%, with SI from 1 to 3) (Figure 1) (Appendix A). 

### 2.2. Binding of In Vitro Growth-Promoting Mechanisms

Grouping these characteristics led to eight groups with 100% similarity (Figure 2). Group I (GI) included two strains that grew without N and produced IAA, while GII also had two strains that, besides the above, also solubilized phosphate. GIII with four strains was like GII but did not grow in N-free media. In comparison, the 10 strains of GIV could only produce IAA, the 19 strains of GV presented all mechanisms, and the 40 strains of GVI had all but P solubilization. GVII is formed by 34 IAA- and siderophore-producing strains, and GVIII by 21 strains with all mechanisms bar growth in N-free media. 

### 2.3. Efficiency of Growth-Promoting Bacteria in the Early Stage of Maize

Generally, the pangolão isolated strains yielded higher results than CI for at least two variables in the N-restricted experiment (Table 1). Although no strain led to significantly higher SDM than CI, several were significantly higher in RECI, indicating an overall higher growth promotion activity. However, all were significantly lower in RE + C than the +C treatment, indicating that the growth promotion could not fully overcome N deficiency effects. 

Although no strain from Pangolão led to higher LA than CI, most other variables had at least two strains with higher results (Table 2). Most importantly, two strains had higher SDM and RECI, while several increased RDM, and four achieved higher RE + C than fully irrigated plants. 

Although no strain significantly differed from CI for any of the variables on the low-Fe solubility experiment (Appendix A), plants with strain 252 were significantly higher, and strain 5227 had a higher SDM. While no strain achieved higher SDM, SAN, or relative efficiencies than CI, several led to significantly higher PH, CD, LA, RDM, and SAP (Table 3). Some had non-significant higher RE + C than +C plants. 

### 2.4. Binding of In Vitro and In Vivo Growth-Promoting Mechanisms

Most in vitro plant-growth-promoting characteristics were not significantly correlated to plant growth effect in maize (Table 4), and even those which were significant, such as SI and PH for the siderophore experiment (r = 0.25, *p* < 0.01), or between SI and SDM in the P solubilization experiment (r = 0.30, *p* < 0.01), had relatively low predictive values. 

As a group, strains that did not grow in N-free media had lower leaf area in the N-deficient experiment and increased RDM in the low-solubility Fe one (Table 5). At the same time, those positive for this mechanism reduced SAN and SAP in the water-deficit experiment. While low or average IAA synthesis led to generally higher results in the water-deficit experiment, those with high IAA synthesis increased RDM in the low-solubility P experiment. Medium SI strains had generally higher RDM in the low-solubility P experiment, while this was found for the high SI strains in the low Fe solubility experiment. These results, coupled with the low and mostly non-significant correlations between in vitro and maize results, indicate that these in vitro methods are not good predictors of plant effects.

## 3. Discussion

This is the first evaluation of the plant-growth-promoting potential of Pangolão grass endophytic bacteria, which is particularly interesting since endophytic bacteria from plants from stressful environments such as this might help reduce environmental stress effects on crops [5,18]. Strains previously isolated from this plant species [13] included several genera known to harbor plant-growth-promoting strains. Indeed, all 132 strains tested under in vitro conditions here had at least one commonly held plant-growth-promoting trait.

These traits occurred in different proportions when other plant species were evaluated. For example, while 48% of the strains grew in N-free media, strains from other grasses grown in the Brazilian tropical semiarid included 66% diazotrophs [20], and signal grass root and rhizosphere bacteria had 58% diazotrophs [21]. 

IAA production, on the other hand, seems to be much more widely spread, since 100% of the strains were positive for this trait, as also happened with strains from cold-tolerant rice [22,23], and strains from plants under saline conditions also had 86% occurrence for this trait [24]. A similarly frequent feature seems to be siderophore production since we found 86% of siderophore-producing strains and signal grass had very similar results at 84% occurrence [25].

Most Pangolão-isolated strains had at least one of the commonly evaluated in vitro growth-promoting characteristics [26], while strains with more than one of these tended to have stronger plant-growth-promoting effects under environmental stress [8]. This may indicate that these bacteria are an important part of Pangolão adaptation mechanisms to the environmental stress typical of where it was collected. This is furthered by the presence of *Pantoea*, *Enterobacter*, *Rhizobium*, *Pseudomonas*, and *Stenotrophomonas* strains, which are frequently described as promoting plant growth for several species [14,27,28,29,30,31,32].

The high proportion of in vitro plant growth characteristics found here agree with the frequent consideration that plants adapted to stressful environments tend to have plant-growth-promoting endophytic bacteria [8,26,33]. Although we evaluated only the initial maize growth, several strains increased some of the plant variables found in plants receiving a CI with known effective strains for this crop under low N availability [17]. This is likely due to IAA effects on root growth, and thus on water and nitrogen absorption [2].

Interestingly, the strains with higher N accumulation in the low-water-availability experiment were all identified as *Rhizobium*, *Paenibacillus*, and *Agrobacterium* species, all of which are known as potentially diazotrophic [34,35], which may indicate some level of unmeasured biological nitrogen fixation.

On the other hand, the small effects of both low P and Fe solubility experiments might be due to the early stage on which we harvested the maize, as also observed for wheat [32]. Still, even under those limiting conditions, some strains induced higher growth than the CI.

A common occurrence in our experiments was bacteria with low values for a given in vitro mechanism achieving higher than CI values for a plant-growth promotion in the experiment designed to evaluate that trait. This is coupled with a generally low and non-significant correlation between in vitro and maize effects and might be due to most of the strains presenting more than one of the mechanisms (Figure 2), as also proposed in the literature and in the typical indication of using a mix of strains for inoculant production [36,37,38,39,40].

## 4. Materials and Methods

Plant sampling, bacterial sampling, isolation, DNA extraction, BOX-PCR grouping, and sampling were all described in Alves et al. [13]. In short, endophytic and rhizospheric bacteria from Pangolão grass from three Pernambuco state municipalities were isolated, through serial dilutions in sterile saline solution, in semi-solid, N-free, NFB media [41], while in one of the locations, JNFB and JMV media were also used [42,43]. Where growth was observed in the N-free media, bacteria were transferred to YMA media for phenotypical characterization [44], followed by DNA extraction and BOX-PCR fingerprinting. Representative strains of the BOX-PCR 90% similarity groups had their 16S rRNA sequenced. They were chosen for the in vitro evaluation of biological nitrogen fixation [41], IAA production [45], calcium phosphate solubilization [46], and siderophore production [47]. 

Each of these mechanisms was later evaluated in a separate greenhouse experiment with maize to evaluate specific environmental stresses, such as low available nitrogen, water deficit, and low-solubility P and Fe sources, respectively.

All experiments were conducted in 5 L plastic vessels filled with sterile sand: vermiculite 1:1 mixture, using maize cultivar BR-5026, also known as São José. This cultivar is recommended for low-resource agriculture in tropical semiarid regions and was developed as a multipurpose cultivar for forage production, immature corn, or for grain production. It has dented yellow cobs, 300 cm tall plants, a 120-day cycle, and a stable yield. 

Seeds were surface-disinfested with 70% ethanol for 30 s, immersed in 2.5% sodium hypochlorite for 2 min, washed eight times with sterile distilled water, and inoculated with 1 mL per seed of 10^9^ cells.mL^−1^ bacterial broth of the strain under evaluation, according to standard Brazilian recommendations for the commercial inoculation of corn [17]. All experiments were conducted on a randomized block design with three replicates.

As biological nitrogen fixation is not considered to support all plant growth for non-legumes [48], all plants in this experiment were supplied with 30% of the recommended N dose at seeding. Twenty strains were evaluated; seven were positive for N-free media growth, and the remainder did not grow in this media (Table 6). The control treatments for this experiment were non-inoculated plants inoculated with the commercial *Azospirillum brasilense* inoculant AzzoFix^®^ (CI). This is a liquid inoculant for corn with strains Ab-V5 and Ab-V6 and 2.0 × 10^8^ UFC·mL^−1^, as defined and authorized according to Brazilian legislation [17], and 100% N dose supply (equivalent to 20 kg N·ha^−1^) (+C) under a low N input recommendation for the tropical semi-arid condition in Pernambuco state [49].

IAA was evaluated under water deficit stress since this mechanism is supposed to increase root growth, maintained by keeping the plants at *ca.* 30% pot-water-holding capacity after complete seed germination. Twenty-four strains were selected representing the 0–5, 5–25, 25–50, 50–75, 75–95, and 95–100 percentiles for in vitro IAA production (Table 6), with the same control treatments as above, except for the positive control substituted for the use of 100% pot-water holding capacity (+C).

For the P solubilization capacity, tri-calcium phosphate was used as the sole P source, and 21 bacterial strains were selected to represent the 0–75, 75–95, and 95–100 percentiles for in vitro P solubilization, respecting the proportion of non-solubilizing bacteria effectively found similarly to the BNF experiment (Table 6). Control treatments were similar to the previous experiments, with the positive control being triple superphosphate as the phosphorus source (+C). 

Siderophore was evaluated by replacing the Fe source in the nutrient solution with a low-solubility one. In this case, 24 strains were selected to represent the 0–5, 5–25, 25–50, 50–75, 75–95, and 95–100 percentiles for in vitro siderophore production (Table 6), and the control treatments were similar to the previous experiments, with the positive control being a high-solubility Fe source (+C).

For all experiments, harvest was 20 days after emergence, plant height (PH), colm diameter (CD), and leaf area (LA) were measured, and root (RDM) and shoot dry matter (SDM) was determined. Traditional wet methods determined N and P contents, and shoot accumulated N (SAN) and P (SAP) were calculated by the product of shoot dry matter and the appropriate content.

We also calculated the relative efficiencies of each treatment in relation to the non-inoculated control (REN), commercial inoculant (RECI), and each positive control (RE + C). In all cases, these were calculated according to this equation.
(1)$$RE\left(\%\right)=\frac{Treatment\;Shoot\;Dry\;Matter}{Average\;Shoot\;Dry\;Matter\;of\;the\;appropriate\;control}\times 100\;\;\;\;\;\;\;\;\;\;\;\;$$

The in vitro data were grouped at 100% similarity using the Jaccard dissimilarity index and UPGMA. Maize data were evaluated for homoscedastic and outliers. When needed, they were transformed by log_10_ and elimination of outliers before ANOVA and Dunnett’s test at 5% significance using the commercial inoculant as the control treatment. Later, only the inoculant treatments were evaluated, considering their grouping from the in vitro evaluations, as described for each experiment, and considering the variation within each group as a random effect. The Pearson linear correlation was evaluated for each experiment, considering the in vitro values and the plant variables.

## 5. Conclusions

Endophytic bacteria strains from pangolão grass increased initial plant growth for at least some variables of those provided by known effective *Azospirillum* strains under four different environmental stress conditions, likely due to this plant being adapted to stressful environments. Thus, its microbiome is an essential component of this adaptation.

The lack of substantial predictive value for the in vitro plant-growth-promotion characteristics, as seen by the low and generally non-significant correlations and the apparent disconnect between these and plant effects, indicates that the common practice of choosing strains for plant evaluation based on the in vitro evaluation might be flawed and needlessly reduce the genetic diversity in the plant evaluation. 

Although the maize experiments were designed to evaluate specific environmental stresses purportedly linked to each in vitro plant-growth-promoting characteristic, the plant promotion is not directly related to individual mechanisms.

## Figures and Tables

**Figure 1 plants-12-02589-f001:**
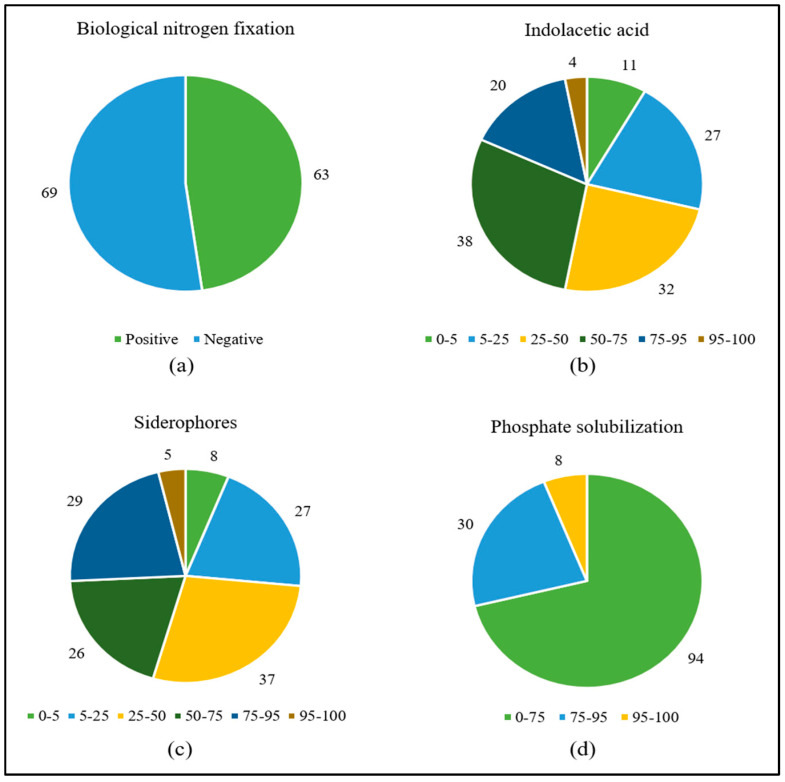
Number of pangolão grass endophytic bacteria positive for growth in N-free media (**a**) or presenting given percentiles of IAA synthesis (**b**), siderophore production (**c**), and calcium phosphate solubilization (**d**).

**Figure 2 plants-12-02589-f002:**
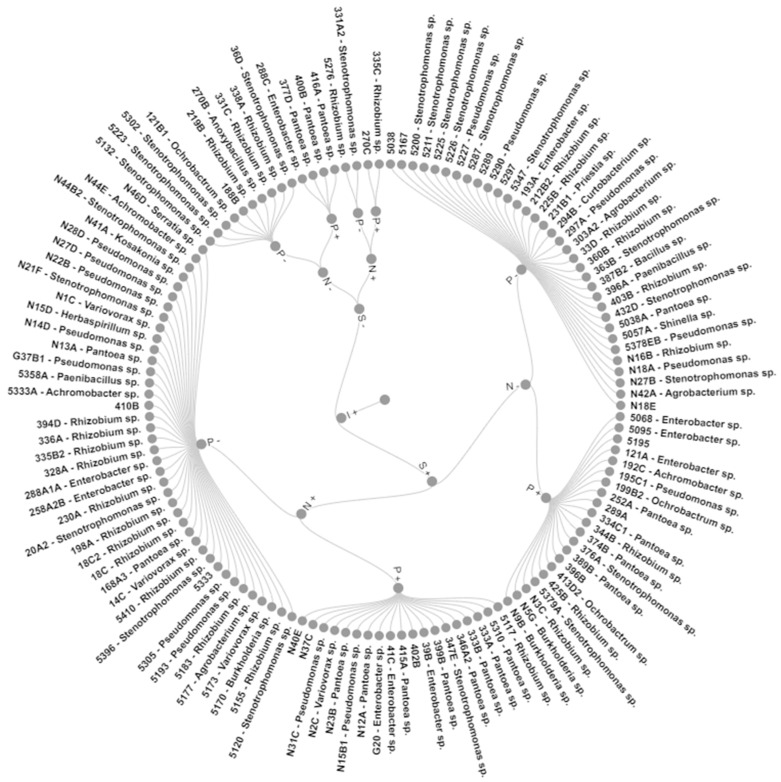
Pangolão grass endophytic bacteria grouped by in vitro growth promotion mechanisms. N—growth in N-free media; I—IAA production; S—siderophore production; P—calcium phosphate solubilization; +—bacteria presents the trait; −—bacteria does not present the trait.

**Table 1 plants-12-02589-t001:** Growth performance of maize inoculated with pangolão endophytic bacteria under restricted N availability.

Treatments	PH (cm)	CD (mm)	LA	SDM	RDM	SAN	SAP	RECI	REN	RE + C
			(cm^2^)	(mg)		mg Plant^−1^			%	
14C	73 *	4.8	82 *	1029	788	87	8	198 *	79	56
212B2	65 *	5.3	77 *	1113	655 *	89	8	218 *	86	61
252A	65 *	5	94	895	506 *	90	9	172	69	49
333B	61	4.8	82 *	759	528 *	56	8	181	58	41
338A	60	4	87 *	861	436	87	9	166	66	47
36D	67 *	5.1	91 *	817	566 *	94	7	157	63	44
389B	65 *	5.7	98 *	661	583 *	61	5	127	51	36
394D	72 *	5.5	91 *	994	668 *	85	8	191 *	76	54
396B	71 *	5.9 *	116	968	563	85	7	186 *	75	53
432D	71 *	5.4	97 *	1078	759	99	7	207 *	83	59
5038	68 *	5	70	890	694 *	79	7	171	69	48
5038A	64	4.8	103	856	745 *	68	9	166	66	47
5057A	66 *	5.4	98	911	524 *	65	8	175	70	50
5211	69 *	5.7	107	1101	766	107	8	212 *	85	60
5297	67 *	5.3	108	838	706 *	73	7	161	64	46
5347	72 *	5.8	123	1073	501 *	100	11 *	206 *	83	58
5358A	68 *	5.3	91 *	939	667 *	86	7	181	72	51
N1C	65 *	4.6	92	894	674 *	87	8	172	69	49
N27D	63	5.1	85 *	874	568 *	78	7	168	67	48
N37C	62	4.5	79 *	668	543 *	88	3	124	51	36
+C	73 *	6.7 *	128	1786 *	1543	158 *	19	274 *	137 *	97 *
NI	70 *	5	99 *	1261 *	867	108	10 *	242 *	97 *	69 *
CI	53	4.5	48	498	225	61	5	93	38	27
CV	1.7	5.9	3	3.5	3.9	4.8	14.5	3.9	5.6	6.1
SDE	0.03	0.04	0.06	0.10	0.11	0.09	0.13	0.09	0.10	0.10

+C = Application of 100% of the recommended dose; NI = no inoculation; CI = commercial inoculant; PH = plant height; CD = colm diameter; LA = leaf area; SDM = shoot dry mass; RDM = dry mass of the root system; SAN = accumulation of N in the aerial part; SAP = accumulation of P in the aerial part; RECI = efficiency relative to the commercial inoculant; REN = non-inoculated control relative efficiency; RE + C = relative efficiency positive control. CV = coefficient of variation. SDE = residual standard deviation. RDM, SDM, SAN, SAP, RER, REN, and RE + C data were log10 transformed. Means followed by an asterisk differ significantly from inoculation with the recommended strain at 0.05 probability by Dunnett’s test.

**Table 2 plants-12-02589-t002:** Growth performance of maize inoculated with pangolão endophytic bacteria under restricted water availability.

Treatments	PH (cm)	CD (mm)	LA	SDM	RDM	SAN	SAP	RECI	REN	RE + C
			(cm^2^)	(mg)		mg Plant^−1^			%	
121B1	58	4.5	66	556	891	37	6 *	97	101	102
192C	60	4.1	73	564	664	46	5	100	104	106
212B2	59	4.3	76	560	866	36	4	99	104	105
230A	65	4.3	74	718	595	35	3	127	133	134
289A	60	4.4	75	465	579	37	4	83	86	87
303A2	59	4.6	74	626	813	46	6 *	109	114	115
333B	65	4.9	85	702	613	38	4	125	130	131
335C	71 *	5.4 *	74	642	803	59 *	6	114	119	120
338A	60	6.5	106	727	939 *	86 *	5 *	108	113	114
402B	60	5.2 *	72	700	1066 *	123	7 *	124	130	131
415A	56	4.9	53	614	798	41	4	109	114	115
425B	51	4.4	64	496	993 *	57 *	3	88	92	93
432D	60	4.2	76	585	657	33	2	104	108	109
5038	61	5	100	688	898	34	3	122	127	129
5038A	64	4.6	104	1140 *	1450	43	4	202 *	211 *	213 *
5095	56	4.1	49	455	911 *	32	4	78	81	82
5155	58	3.5	70	499	784	43	5 *	92	96	97
5211	63	4.2	76	572	766	40	3	101	106	107
5276	60	4.5	81	659	803	43	4	145 *	151 *	153 *
5287	65	4.6	58	633	750	36	4	112	117	119
5289	67 *	4.9	72	804	859	44	4	140 *	146 *	148 *
5347	63	5.1	78	661	907	49	5	117	122	124
5358A	68 *	4.9	81	821 *	1010 *	77 *	8	146 *	152 *	154 *
+C	54	4.6	55	470	261	77 *	6 *	80	84	85
N42A	60	4.2	77	579	670	61 *	3	103	107	108
NI	56	4.7	74	505	1149 *	46	4	72	75	76
CI	55	4	74	432	384	28	3	75	79	79
CV	1.7	6.2	4.1	3.4	3.8	6.2	14.4	4	4	4
SDE	0.03	0.04	0.08	0.10	0.11	0.10	0.09	0.08	0.08	0.08

+C = 100% pot-water holding capacity; NI = no inoculation; CI = commercial inoculant; PH = plant height; CD = colm diameter; LA = leaf area; SDM = shoot dry mass; RDM = dry mass of the root system; SAN = accumulation of N in the aerial part; SAP = accumulation of P in the aerial part; RECI = efficiency relative to the commercial inoculant; REN = non-inoculated control relative efficiency; RE + C = relative efficiency positive control. CV = coefficient of variation. SDE = residual standard deviation. Height, LA, SDM, RDM, SAN, and SAP data were transformed in to log10. Means followed by an asterisk differ significantly from inoculation with the recommended strain at 0.05 probability by Dunnett’s test.

**Table 3 plants-12-02589-t003:** Growth performance of maize inoculated with pangolão endophytic bacteria with low solubility P.

Treatments	PH (cm)	CD (mm)	LA	SDM	RDM	SAN	SAP	RECI	REN	RE + C
			(cm^2^)	(mg)		mg Plant^−1^		%		
252A	76	5.6	120	1318	932	61	9	140	140	146
333B	86 *	5.6	157 *	1315	1177	95	9	140	139	146
334C1	81	6.1	134	1558	1339	125	13	165	165	173
335C	74	7.4 *	123	1565	1441 *	138	9	166	166	174
347B	71	5.2	104	1002	1220	107	13 *	106	106	111
377D	79	6.6	147	1655	2791	87	7	176	175	184
389B	55	5.3	106	538	642	119	15 *	109	109	114
396B	90 *	6.1	118	1448	1026	71	6	154	153	161
413D2	77	5.8	100	1370	2131 *	97	8	145	145	152
41C	79	5.7	121	1300	1265	90	10	138	138	144
432D	79	5.4	101	1298	933	116	8	138	138	144
5038	74	6.2	121	1275	593	100	8	135	135	142
5057A	96 *	6.5	139	1966	1631 *	106	15 *	209	208	218
5095	72	5.4	106	1031	1029	82	7	109	109	115
5211	79	5.5	117	1123	1117	67	7	119	119	125
5347	73	5.1	107	1173	812	111	8	128	128	134
G20	74	5.7	102	1384	1413 *	119	16 *	147	147	154
N37C	75	7 *	168 *	1359	1564 *	122	15 *	144	144	151
N40E	71	5.6	88	1227	1595 *	112	9	130	130	136
N5G	87	7.3	126	1714	1510 *	163	15 *	182	182	190
N9B	76	6.2	122	1305	1016	88	9	139	138	145
+C	66	5.2	82	876	2016 *	69	6	93	93	97
NI	79	5.6	114	943	911	81	8	100	100	105
CI	66	5	95	931	614	87	5	99	99	103
CV	2	6.6	3.6	5.4	4.4	8.4	14.5	6.3	6.3	6.3
SDE	0.04	0.05	0.07	0.16	0.14	0.17	0.14	0.14	0.14	0.14

+C = recommended dose of P_2_O_5_ provided by triple superphosphate; NI = no inoculation; CI = commercial inoculant; PH = plant height; CD = colm diameter; LA = leaf area; SDM = shoot dry mass; RDM = dry mass of the root system; SAN = accumulation of N in the aerial part; SAP = accumulation of P in the aerial part; RECI = efficiency relative to the commercial inoculant; REN = non-inoculated control relative efficiency; RE + C = relative efficiency positive control. CV = coefficient of variation. SDE = residual standard deviation. All data were transformed by log10. Means followed by an asterisk differ significantly from the commercial inoculant control at 0.05 probability, by Dunnett’s test.

**Table 4 plants-12-02589-t004:** Pearson’s correlation between in vitro growth promoting characteristics of the strains and results obtained in the BNF, IAA, siderophore, and phosphate solubilization experiments.

	PH	CD	LA	SDM	RDM	SAN	SAP	RECI	REN	RE + C
BNF										
SI	−0.242	−0.125	−0.093	−0.358 **	−0.173	−0.214	−0.297 *	−0.218	−0.358 **	−0.358 **
IAA	−0.015	0.169	−0.154	−0.096	−0.110	−0.172	−0.091	−0.058	−0.096	−0.096
Siderophore	0.182	0.228	0.143	0.078	0.100	−0.133	0.040	0.113	0.078	0.078
BNF	−0.091	−0.204	−0.303 *	−0.095	−0.048	−0.069	−0.155	0.024	−0.095	−0.095
IAA										
SI	−0.134	0.012	−0.128	−0.090	−0.132	0.018	−0.100	−0.105	−0.105	−0.105
IAA	−0.136	−0.180	−0.072	−0.115	−0.076	−0.013	−0.049	−0.115	−0.115	−0.115
Siderophore	−0.086	−0.205	−0.134	−0.104	−0.115	−0.155	−0.169	−0.139	−0.139	−0.139
BNF	0.068	0.053	0.043	0.059	0.037	0.102	0.061	0.064	0.064	0.064
Siderophore										
SI	0.259 *	0.073	0.134	0.201	0.010	0.028	0.104	0.201	0.201	0.201
IAA	0.160	0.081	0.122	0.216	0.094	0.027	0.130	0.216	0.216	0.216
Siderophore	0.145	−0.045	−0.132	0.058	−0.003	−0.051	−0.149	0.058	0.058	0.058
BNF	−0.008	−0.019	0.073	0.002	0.062	−0.043	0.033	0.002	0.002	0.002
Phosphate Solubilization										
SI	−0.167	0.096	−0.073	−0.009	0.309 *	0.029	0.072	−0.047	−0.047	−0.047
IAA	−0.083	−0.165	−0.212	−0.015	0.068	−0.014	0.094	−0.075	−0.075	−0.075
Siderophore	−0.121	−0.221	−0.159	−0.126	−0.229	−0.080	−0.028	−0.167	−0.167	−0.167
BNF	−0.084	0.119	0.029	0.055	0.139	0.126	0.136	0.007	0.007	0.007

All data were transformed by log10. * and ** significant at 5% and 1% probability, respectively. SI = phosphate solubilization index. PH = plant height; CD = colm diameter; LA = leaf area; SDM = shoot dry mass; RDM = dry mass of the root system; SAN = accumulation of N in the aerial part; SAP = accumulation of P in the aerial part; RECI = efficiency relative to the commercial inoculant; REN = non-inoculated control relative efficiency; RE + C = relative efficiency positive control.

**Table 5 plants-12-02589-t005:** Pangolão endophytic bacteria, grouped according to their in vitro growth promotion mechanisms performance, and their effects on maize growth under different environmental stresses, reduced N availability, water deficit stress, low P solubility, and low Fe solubility.

	PH	CD	LA	SDM	RDM	SAN	SAP	RECI	REN	RE + C
	BNF									
BNF										
+	66a	5a	86b	871a	628a	80a	7a	172a	67a	47a
−	67a	5a	97a	919a	607a	83a	8a	177a	71a	50a
SI										
0–75	68a	5a	94a	928a	631a	83a	8a	179a	71a	50a
75–95	63a	5ab	88a	824a	517a	71a	8ab	177a	63a	45a
95–100	62a	4b	80a	668a	541a	88a	3b	124a	51a	36a
IAA										
5–25	66a	5a	90a	919a	643a	81a	8a	177a	71a	50a
25–50	67a	5a	99a	899a	613a	87a	7a	172a	69a	49a
50–75	66a	5a	90a	850a	582a	75a	7a	170a	65a	46a
75–90	69a	5a	84a	1054a	661a	87a	8a	204a	81a	57a
Siderophore										
0–5	64a	5a	89a	839a	498a	90a	8a	161a	65a	46a
5–25	66a	5a	91a	908a	584a	76a	7a	175a	70a	49a
25–50	66a	5a	92a	808a	590a	83a	6a	154a	62a	44a
CV	1.7	5.9	2.9	3.4	3.9	4.6	14.4	3.9	5.4	5.9
SDE	0.03	0.03	0.06	0.10	0.11	0.09	0.12	0.09	0.10	0.10
IAA										
BNF										
+	60a	5a	75a	614a	832a	43b	4b	108a	112a	113a
−	63a	5a	73a	664a	795a	53a	5a	122a	127a	128a
SI										
0–75	61a	5a	76a	662a	846a	47a	4a	118a	123a	124a
75–95	59a	4a	74a	550ab	696a	44a	4a	98b	102b	103b
95–100	63a	5a	60a	534b	857a	43a	5a	94b	98b	99b
IAA										
5–25	63a	5a	90a	831a	1101a	79a	6a	141a	147a	149a
25–50	64a	5a	76a	611b	778b	44bc	4b	109ab	114ab	115ab
50–75	59ab	5a	68a	605b	796ab	36c	4b	107ab	111ab	112ab
75–90	62ab	4b	70a	630b	749b	38bc	4b	112ab	116ab	118ab
Siderophore										
0–5	63a	5a	80ab	630a	876a	57a	6a	106a	111a	119a
5–25	62a	5a	73ab	683a	880a	59a	6a	126a	132a	108a
25–50	58a	4b	67ab	580a	754b	41b	4b	103a	108a	97a
50–75	61a	4b	76ab	570a	763b	45ab	3b	101a	106a	85a
75–95	63a	5a	85a	717a	844a	41b	4b	127a	132a	103a
95–100	58a	4b	60b	505a	779b	38b	4b	88a	92a	88a
CV	1.7	6	4	3.2	3.8	5.8	14	4.1	4.1	4.1
SDE	0.03	0.04	0.08	0.09	0.11	0.10	0.09	0.09	0.09	0.09
Siderophore										
BNF										
+	73a	6a	108a	1273a	827a	98a	8a	87a	99a	117a
−	72a	6a	113a	1299a	903a	94a	8a	88a	101a	120a
SI										
0–75	71b	6a	107b	1211b	807b	94a	8a	82b	94b	112b
75–95	78a	6a	109ab	1458a	1012a	110a	10a	99a	113a	134a
95–100	76a	6a	118a	1384ab	814ab	98a	8a	94ab	107ab	128ab
IAA										
5–25	68b	6a	106ab	1097b	693a	88a	8a	75b	85b	101b
25–50	75a	6a	115a	1287ab	849a	97a	9a	87ab	100ab	119ab
50–75	75a	6a	108ab	1348a	896a	103a	9a	92a	105a	124a
75–90	72ab	6a	114ab	1433a	888a	98a	8a	97a	111a	132a
Siderophore										
0–5	69b	6a	113ab	1111b	692b	94a	9ab	76b	86b	102b
5–25	72b	6a	113ab	1258ab	966a	93a	8a	86ab	98ab	116ab
25–50	80a	6a	119a	1540a	1017a	123a	10a	105a	119a	142a
50–75	72b	6a	102b	1277ab	835ab	89a	7b	87ab	99ab	118ab
75–95	72b	6a	108ab	1253ab	752ab	96a	10a	85ab	97ab	116ab
95–100	74ab	6a	99b	1259ab	819ab	91a	7b	86ab	98ab	116ab
CV	1.7	6	2.3	2.3	3.5	5.9	10.7	3.7	3.6	3.5
SDE	0.03	0.05	0.05	0.07	0.10	0.12	0.10	0.07	0.07	0.07
Phosphate Solubilization										
BNF										
+	77a	6a	117a	1267a	1145a	97a	9a	141a	140a	147a
−	76a	6a	124a	1354a	1401b	111a	11a	144a	143a	150a
SI										
0–75	77a	6a	115a	1183a	916b	96a	9a	138a	138a	144a
75–95	80a	6a	129a	1389a	1378ab	100a	11a	148a	147a	154a
95–100	74a	6a	113a	1311a	1414a	107a	10a	139a	139a	146a
IAA										
5–25	81a	7a	124a	1496a	1239ab	120a	12a	159a	158a	146a
25–50	75a	6a	122a	1209a	1170b	101a	9a	139a	139a	166a
50–75	78a	6a	120a	1307a	1151b	95a	10a	139a	139a	145a
75–90	75a	6a	101a	1377a	1735a	107a	11a	146a	146a	153a
Siderophore										
0–5	74a	7a	123a	1565a	1441a	138a	9a	166a	166a	174a
5–25	81a	6a	130a	1501a	1651a	105a	12a	159a	159a	167a
25–50	73a	6a	117a	1143a	1151a	98a	10a	135a	134a	141a
50–75	79a	5a	117a	1123a	1117a	67a	7a	119a	119a	125a
75–95	79a	6a	115a	1348a	1102b	107a	10a	144a	143a	150a
95–100	74a	6a	113a	1160a	1022b	85a	8a	123a	123a	129a
CV	1.9	6.8	3.4	5.3	4.1	8.3	13.2	6.1	6.1	6.1
SDE	0.04	0.05	0.07	0.17	0.13	0.17	0.13	0.13	0.13	0.13

Means followed by the same letter do not differ statistically from each other, lowercase in the column, at 0.05 probability, by Tukey’s test. PH = plant height; CD = colm diameter; LA = leaf area; SDM = shoot dry mass; RDM = dry mass of the root system; SAN = accumulation of N in the aerial part; SAP = accumulation of P in the aerial part; RECI = efficiency relative to the commercial inoculant; REN = non-inoculated control relative efficiency; RE + C = relative efficiency positive control. CV = coefficient of variation. SDE = residual standard deviation.

**Table 6 plants-12-02589-t006:** Strains selected for the maize growth-promotion experiments. BNF—reduced N availability, IAA—water deficit stress, P—low P solubility, and Siderophore—low Fe solubility. Grey cells indicate those in common to all experiments. + and - indicate presence or absence of the trait, respectively.

Strain	Classification	BNF	IAA	P	Siderophore
5038		+	+	+	+
5095	*Enterobacter*	-	+	+	+
5155	*Rhizobium*	-	+	-	-
5211	*Stenotrophomonas*	+	+	+	+
5227	*Pseudomonas*	-	-	-	+
5276	*Rhizobium*	-	+	-	-
5287	*Stenotrophomonas*	-	+	-	+
5289		-	+	-	-
5297	*Rhizobium*	+	-	-	+
5347	*Stenotrophomonas*	+	+	+	+
5410	*Rhizobium*	-	-	-	+
121B1	*Ochrobactrum*	-	+	-	+
14C	*Variovorax*	+	-	-	-
192C	*Achromobacter*	-	+	-	+
212B2	*Rhizobium*	+	+	-	+
230A	*Rhizobium*	-	+	-	-
231B1	*Priestia*	-	-	-	+
252A	*Pantoea*	+	-	+	+
289A		-	+	-	-
303A2	*Agrobacterium*	-	+	-	-
331C	*Rhizobium*	-	-	-	+
333B	*Pantoea*	+	+	+	+
334C1	*Pantoea*	-	-	+	-
335C	*Rhizobium*	-	+	+	+
338A	*Rhizobium*	+	+	-	+
344B	*Rhizobium*	-	-	-	+
347B	*Stenotrophomonas*	-	-	+	-
36D	*Stenotrophomonas*	+	-	-	-
377D	*Pantoea*	-	-	+	-
389B	*Pantoea*	+	-	+	-
394D	*Rhizobium*	+	-	-	-
396B		+	-	+	+
402B		-	+	-	-
413D2	*Ochrobactrum*	-	-	+	+
415A	*Pantoea*	-	+	-	-
41C	*Enterobacter*	-	-	+	-
425B	*Rhizobium*	-	+	-	-
432D	*Stenotrophomonas*	+	+	+	+
5038A	*Pantoea*	+	+	-	-
5057A	*Shinella*	+	-	+	+
5358A	*Paenibacillus*	+	+	-	+
G20	*Enterobacter*	-	-	+	-
N1C	*Variovorax*	+	-	-	-
N27D	*Pseudomonas*	+	-	-	-
N37C		+	-	+	-
N40E		-	-	+	-
N42A	*Agrobacterium*	-	+	-	-
N5G	*Burkholderia*	-	-	+	-
N9B	*Burkholderia*	-	-	+	+

## Data Availability

Data are available upon request to contact author.

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
