# Peer review of "Screening Digitaria eriantha cv. Suvernola Endophytic Bacteria for Maize Growth Promotion"

_plants, 2023, doi:10.3390/plants12142589_

Round 1

Reviewer 1 Report

The paper entitled ‘Maize growth promotion by Digitaria eriantha cv Suvernola   endophytic bacteria’endophytic bacteria in maize growth promotion.  I found this study very interesting and informative.. However, I have following comments that needs to be addressed

I suggest authors to modify the tittle. Authors should mention isolation or screening in the tittle.

Abstract is well framed.

I am surprised with introduction section as authors provided little information on the background of the study. I strongly suggest authors to provide more information on the role of endophytes in growth promotion in different crop systems.

Results section: Authors didn’t mention how they screening of endophytes.

Discussion is the weakest point of this good study. I suggest authors to provide more insights on results and discussion

Material and methods can be improved.

Figure legends needs more information

There are some minor grammatical errors that needs to be corrected.

I recommend this paper for publication after addressing above comments

Author Response

We thank the reviewer for the comments, which greatly improved our manuscript. We have attempted to correct all points, as described in the attachment. 

Reviewer 2 Report

Dear authors!

Thank you for submitting the manuscript.

The manuscript is devoted to the now popular topic of the search for microorganisms - stimulators of plant growth and development.

Your article uses many modern and classical methods for studying the properties of microorganisms and their effect on plants. A lot of work has been done.

I have a number of comments:

In the methodical part, you write "inoculated with 1 mL per seed of 109 cells. ml-1 bacterial broth of a strain under evaluation". Please clarify why you chose such a bacterial load on seeds, how does such a titer of bacteria correlate with the density of bacteria in the natural environment?

You write that maize cultivar BR-5026, also known as São José. Please add brief information about this variety.

Explain the choice of the commercial drug "AzzoFix".

Explain why pangolon grass was chosen as the source of endophytic microorganisms.

It is required to carry out work on statistical data processing. In tables 1, 2 and 3, 5, not all data are presented in the form of % in order to show them without deviations. I consider it necessary to add, depending on the nature of the distribution of samples, a standard deviation, a standard error, or the values of the 25th and 75th percentiles. In its present form, the tables look unconvincing. In general, these tables are difficult for the reader to understand. It would be nice to present this data in a different, more visual form.

Negligence in the design of the text and bibliography. For example, in the reference, year 17 is not bolded. There are many hyperlinks in the bibliography that needed to be removed.

Respectfully yours, reviewer

June 09, 2023

Author Response

We thank the reviewer for greatly improving our manuscript with these comments. We have accepted all points, as described in the attached file
